# Buckypapers in Polymer-Based Nanocomposites: A Pathway to Superior Thermal Stability

**DOI:** 10.3390/nano15141081

**Published:** 2025-07-11

**Authors:** Johannes Bibinger, Sebastian Eibl, Hans-Joachim Gudladt, Philipp Höfer

**Affiliations:** 1Institute of Lightweight Engineering, University of the Bundeswehr Munich, 85577 Neubiberg, Germany; philipp.hoefer@unibw.de; 2Bundeswehr Research Institute for Materials, Fuels and Lubricants, 85435 Erding, Germany; sebastianeibl@bundeswehr.org; 3Institute of Materials Science, University of the Bundeswehr Munich, 85577 Neubiberg, Germany; hans-joachim.gudladt@unibw.de

**Keywords:** carbon fiber-reinforced polymer (CFRP), buckypaper, carbon nanotube (CNT), improved heat resistance, prediction model for thermal properties

## Abstract

The thermal stability of carbon fiber-reinforced plastic (CFRP) materials is constrained by the low thermal conductivity of its polymer matrix, resulting in inefficient heat dissipation, local overheating, and accelerated degradation during thermal loads. To overcome these limitations, composite materials can be modified with buckypapers—thin, densely interconnected layers of carbon nanotubes (CNTs). In this study, sixteen 8552/IM7 prepreg plies were processed with up to nine buckypapers and strategically placed at various positions. The resulting nanocomposites were evaluated for manufacturability, material properties, and thermal resistance. The findings reveal that prepreg plies provide only limited matrix material for buckypaper infiltration. Nonetheless, up to five buckypapers, corresponding to 8 wt.% CNTs, can be incorporated into the material without inducing matrix depletion defects. This integration significantly enhances the material’s thermal properties while maintaining its mechanical integrity. The nanotubes embedded in the matrix achieve an effective thermal conductivity of up to 7 W/(m·K) based on theoretical modeling. As a result, under one-sided thermal irradiation at 50 kW/m^2^, thermo-induced damage and strength loss can be delayed by up to 20%. Therefore, thermal resistance is primarily determined by the nanotube concentration, whereas the arrangement of the buckypapers affects the material quality. Since this innovative approach enables the targeted integration of high particle fractions, it offers substantial potential for improving the safety and reliability of CFRP under thermal stress.

## 1. Introduction

Carbon fiber-reinforced polymers (CFRP) are indispensable in modern aviation [1,2]. Their superior specific stiffness and strength make them the material of choice for critical components like wings, fuselages, and structural panels in aircraft [3,4]. During flight, these parts can be exposed to various thermal loading scenarios, ranging from moderate heat fluxes, such as heating through daily solar radiation [5] or turbine walls [6], to extreme conditions like fully developed fires [5,7,8,9] or lightning strikes [10]. However, compared to alternative lightweight metals, these composites are limited by the relatively low thermal stability of their polymer matrix [11,12,13,14]. For instance, the widely utilized 8552 matrix is engineered to withstand temperatures of only up to 121 °C [15]. Under certain conditions, this threshold can be quickly exceeded, leading to thermo-induced damage and a loss of mechanical integrity [16,17,18].

To mitigate the risk of localized overheating, nanoparticulate additives such as carbon nanotubes (CNTs) can be incorporated into the material due to their exceptional potential for high longitudinal thermal conductivity. However, their effective conductivity strongly depends on factors such as structure, morphology, and synthesis methods [19]. Consequently, thermal conductivity can vary widely—from below 0.1 W/(m·K) [20,21] for aggregated multi-walled CNTs up to 6600 W/(m·K) [22] for individual single-walled CNTs at room temperature. The fabrication of these polymer-based nanocomposites can be performed in various ways. In melt processing, CNTs are directly integrated into the matrix and dispersed under mechanical forces. Unfortunately, even at low concentrations, the dynamic viscosity of the resin increases significantly due to the high aspect ratio of the nanotubes, complicating processing and promoting the formation of agglomerates [23,24]. Consequently, the maximum nanotube loading is limited to approximately 1.5–2.0 wt.% in the polymer matrix [23,24]. To improve processability, the solvent-based method can be employed. Hereby, the nanoparticles are first dispersed in an organic solvent before the polymer is dissolved in the same mixture [25]. The solvent is subsequently removed through drying, leaving behind a finely dispersed CNT network in the resin [25]. However, this method only allows for slightly higher mass fractions between 2.0 and 5.0 wt.% nanotubes [26,27,28]. As a result, the thermal conductivity of the nanocomposites improve only modestly by around 10–15% [23,24,28]. This limitation arises from several factors. Firstly, in the absence of an external magnetic field, nanotubes are typically randomly oriented within the composite, which significantly restricts heat transfer [28]. Secondly, the weak van der Waals interactions between CNTs introduce substantial thermal resistance at their interfaces [24,29]. Therefore, under thermal loading, the strength loss is only marginally delayed by about 7% [23,24]. While nanotubes hold considerable potential for enhancing material properties, their random orientation and the low achievable concentrations in the material represent a critical bottleneck.

One promising approach to overcome this limitation is the use of pre-fabricated, thin, dense, yet flexible nanotube network films, known as buckypapers [30]. On the one hand, these films can exhibit effective thermal conductivities ranging from 0.1 up to 770 W/mK, depending on the intrinsic properties of the CNTs, packing density, alignment, and layer structure [19]. On the other hand, they can be directly processed with prepreg plies, enabling an innovative approach for more efficient integration of CNTs into the composite material. As a result, high particle loadings can be incorporated in a targeted and controlled manner without requiring dispersion during processing. Such capability opens up a wide array of new applications in aviation applications for these materials [31,32]. Current research primarily focuses on the effects of buckypapers on electrical properties [33,34,35], lightning protection [36,37,38], electromagnetic interference (EMI) shielding [39,40,41], deicing performance [42,43,44], mechanical properties [45,46,47,48], and flame retardants [49,50], while their impact on the thermal resistance of CFRP composites remains largely unexplored.

To address this gap, this study explores the incorporation of varying numbers and positions of buckypapers within the material. For this purpose, up to nine buckypapers were alternately laminated with sixteen prepreg plies, aiming to strategically place them near the front side, while in a periodic arrangement, they are gradually shifted deeper until a symmetrical configuration achieves a uniform distribution throughout the entire volume. The resulting nanocomposites were evaluated in terms of processing limitations, property modifications, and thermal resistance. The goal is to gain a deeper understanding of the interaction between buckypaper and the composite material, ultimately striving to optimize thermal resilience.

## 2. Materials

The commercially available CFRP HexPly^®^ 8552/IM7 from Hexcel Composites GmbH (Stade, Germany) is a high-performance prepreg material, specifically designed for primary aerospace structures [15]. The HexPly^®^ 8552 matrix consists of epoxy components, including tetraglycidyl 4,4’-diaminodiphenylmethane (TGDDM) and triglycidyl-para-aminophenol (TGPAP), the curing agent 3,3’- or 4,4’-diaminodiphenyl sulfone (DDS), and the thermoplastic toughener polyethersulfone (PES) [51]. The HexTow^®^ IM7 fibers are continuous, intermediate-modulus polyacrylonitrile-based fibers with a filament count of 12 K [52]. As modifying components, buckypapers from NanoTechLabs (Yadkinville, NC, USA) were utilized. These were produced via vacuum filtration of approximately 2 g of multi-walled carbon nanotubes, resulting in buckypapers with dimensions of 300 × 300 × 0.13 mm^3^, an areal weight of 20 g/m^2^, and a porosity of 85–90% [53,54]. The nanotubes themselves exhibit a purity of around 95% [53], with trace amounts of iron residues, likely from ferrocene-based catalysts used in the chemical vapor deposition (CVD) during the manufacturing process [55].

The non-modified material $b_{0}p_{0}$ consists of 16 prepreg plies with a nominal cured ply thickness of 0.131 mm and a fiber orientation of [(+45°/+90°/−45°/0°)_2_]_s_ [15]. To fabricate the nanocomposites, buckypapers were integrated into the material in varying quantities (*b*) and positions (*p*), as shown in Figure 1. In the alternating series, an increasing number of buckypapers ($b_{1}$, $b_{2}$, $b_{5}$, and $b_{9}$) were interleaved with single prepreg plies ($p_{1}$), starting from the front side. In the periodic series, a constant number of five buckypapers ($b_{5}$) was integrated, separated by different numbers of prepreg plies ($p_{1}$, $p_{2}$, and $p_{4}$). In the symmetrical series, five and nine buckypapers ($b_{5}$ and $b_{9}$) were evenly distributed over the cross-section ($p_{4}$ and $p_{2}$).

The laminated stacks were cured in a 13500 L hot air autoclave from Scholz, according to the manufacturer’s specifications [15]. Throughout the process, a vacuum of 0.2 bar was applied in the vacuum bag, while a pressure of 6.5 bar was maintained inside the autoclave. During the first step, the laminate was heated to 80 °C for one hour, followed by a temperature increase to 180 °C for two hours in the second step. After cooling, the laminates were examined for quality defects, such as pores, macroscopic delaminations, or errors in fiber orientation, using the Hill-Scan 3060 UHF ultrasonic testing system from Dr. Hillger (Braunschweig, Germany), and were then approved for further testing.

## 3. Methods

### 3.1. Buckypaper Infiltration

The diffusion process of the polymer matrix into the buckypapers was initially investigated to gain insight into the microstructural behavior of the nanocomposites. Therefore, the Ultra Plus Field Emission Scanning Electron Microscope (SEM) from Zeiss (Oberkochen, Germany) was used with an acceleration voltage of 1 kV, enabling magnification of the individual components and identification of defects, such as delamination caused by matrix depletion within the material. To quantify these defects with high precision down to the 10 μm range, the micro-computed tomography (μCT) system V-TOME XL 300 from General Electric (Frankfurt am Main, Germany), equipped with a 180 kV microfocus X-ray source, was employed in conjunction with grayscale analysis. The resulting fraction of carbon nanotubes within these nanocomposite was determined by the mass ratio of the unprocessed buckypapers to the cured laminate. Their content in the matrix was calculated based on the polymer mass fraction of 0.32 wt.%.

### 3.2. Material Properties

The impact of the infiltrated buckypapers on the functional characteristics was examined to assess the performance of the nanocomposites. For this, the electrical behavior was evaluated by measuring the electrical resistance in both lateral and vertical directions using a sponge rubber electrode in accordance with DIN EN 61340-2-3 [56]. These measurements were conducted under controlled temperature and humidity conditions over a duration of 60 s. For resistance values up to the megohm range, a Fluke multimeter was employed, while a Sefelec megohmmeter was utilized for measurements extending into the teraohm range.

The thermal properties of the materials were analyzed using the Light Flash Analysis (LFA) 467 HAT HyperFlash^®^ from Netzsch (Selb, Germany). Therefore, thermal diffusivity, thermal conductivity, and specific heat capacity were determined in accordance with DIN EN ISO 22007-4 [57]. The material density was measured using a CM224S balance from Sartorius (Göttingen, Germany), following the Archimedean principle as specified in DIN EN ISO 1183-1 [58].

The glass transition temperature of the composites was evaluated via a three-point bending test, using a Gabo Eplexor^®^ 500 dynamic mechanical analyzer (DMA) from Netzsch. During this process, specimens were subjected to a controlled heating rate of 1 °C/min, while experiencing static and dynamic loading at a frequency of 1 Hz.

Mechanical properties were assessed with universal testing machines from Zwick/Roell (Ulm, Germany). Interlaminar shear strength was measured on 20 × 10 × 2 mm^3^ specimens, in accordance with DIN EN 2563 [59], at a traverse speed of 1 mm/min, using supports with a radius of 3 mm and a span of 10 mm. Flexural strength was tested on 100 × 15 × 2 mm^3^ specimens, following DIN EN ISO 14125 [60], at 5 mm/min, using supports with a radius of 2 mm and a span of 81 mm. Compressive strength tests were performed on 110 × 10 × 2 mm^3^ samples, using a Celanese fixture, at 1 mm/min, in accordance with DIN EN ISO 14126 [61]. Finally, tensile strength was determined on 250 × 10 × 2 mm^3^ specimens, at 2 mm/min, following DIN EN ISO 527-4 [62].

### 3.3. Thermal Resistance

To investigate the thermal stability of the modified materials, 20 × 10 × 2 mm^3^ samples were exposed to an electric conical radiant heater within the Cone Calorimeter from Fire Testing Technology (East Grinstead, England). The irradiations were conducted at a heat flux of 50 kW/m^2^ in 5 s intervals, for a total duration of up to 40 s. The temperature evolution was recorded using embedded type K thermocouples (nickel–chromium/nickel) from Thermodirekt (Bruchköbel, Germany) at different ply depths. Thermal damage, such as matrix degradation, was analyzed through infrared spectra in the wavenumber range of 4000 to 400 cm^−1^ using the Tensor 27 from Bruker (Ettlingen, Germany). The system featured a Harrick ATR cell, a silicon crystal, and a pyroelectric detector. Structural damage, including thermo-induced delaminations, was examined using μCT, analogous to the previously discussed defects. Finally, the impact of these thermal damage processes on the interlaminar shear strength was determined under the aforementioned conditions, with the thermally exposed side resting on the two lower supports.

## 4. Results and Discussion

### 4.1. Characteristics of Buckypaper

For the modification of polymer-based composite materials with buckypaper, certain conditions must be met. To investigate these conditions, the morphology of the unprocessed buckypaper is analyzed using the SEM images in Figure 2. The left micrograph (a1), captured at low magnification, clearly illustrates the paper’s microscopic structure as a complex and tightly interconnected network of CNTs. This network arises from the mechanical entanglement of the nanotubes and their mutual attraction through van der Waals forces [63]. As a result of this unique structure, the material acquires exceptional mechanical strength and flexibility, both of which are crucial for effective handling and processing during the infiltration process. The right micrograph (a2) provides a closer inspection of the nanoscale arrangement of the CNTs. Here, the nanotubes appear predominantly randomly oriented, contributing to the anisotropic behavior of the buckypaper in the lateral plane. The gaps between the nanotubes are sufficiently large to allow the polymer matrix to infiltrate during processing.

The nanotubes themselves extend to micrometer-scale lengths, while their diameters remain within the nanometer scale, resulting in an exceptional aspect ratio. To quantify the size distribution, a histogram of the nanotube diameters is presented in Figure 2b. The analysis reveals that the diameters range from 3 to 255 nm, following a Gaussian distribution with a root mean square error (RMSE) of 4.5. Consequently, 68% of the diameters fall within the range of 20.4 ± 8.3 nm. This indicates that the buckypaper is predominantly composed of thinner nanotubes, which offer a significantly larger surface area, enhancing interaction with the polymer matrix in the composite material. Thus, all the essential prerequisites of the buckypaper for the fabrication of high-performance nanocomposites are fulfilled, including mechanical durability and flexibility, macroscale uniformity, and adequate permeability for resin infiltration.

### 4.2. Infiltration Process

The fabrication of the nanocomposites is achieved by infiltrating buckypapers with a polymer matrix. This process begins with precise control of both temperature and pressure inside the autoclave. At a temperature of approximately 80 °C, the viscosity of the pre-cured resin within the prepreg plies is sufficiently reduced, enabling it to flow. Simultaneously, an overpressure of 6.5 bar compresses the plies, promoting not only their consolidation but also the diffusion of the low-viscosity resin into the buckypapers. A vacuum of 0.2 bar is applied in the vacuum bag to prevent any residual air inclusions in the material. Once the infiltration is complete, the epoxy resin is fully cured at 180 °C. The effect of the number and positioning of the buckypapers in the laminate on this process will be explained in detail in the following section.

For the qualitative evaluation of the infiltration process, SEM images from the surface and cross-section of the laminates are presented in Figure 3. In the unmodified reference material $b_{0}p_{0}$ (a1), a wavy polymer structure is visible on the surface, resulting from the imprint of the peel ply used during the laminate curing process in the autoclave. Along the cross-section (a2), the first four prepreg plies (+45°/90°/−45°/0°), with a nominal cured thickness of approximately 115 ± 5 μm, are clearly distinguishable. These plies feature a prominent matrix layer of about 20 ± 5 μm on the surface (a3), while thinner layers of approximately 2 ± 1 μm are observed at their interfaces (a4). Consequently, the 8552/IM7 system demonstrates a moderate matrix excess that can be utilized for the infiltration of the buckypapers.

When integrating up to two buckypapers with an intermediate prepreg ply $b_{2}p_{1}$ (b1), the matrix-bound carbon nanotube structure becomes visible on the surface. These nanotubes predominantly retain their random planar alignment, similar to the unprocessed buckypaper shown in Figure 2a, indicating that the infiltration process does not significantly affect their surface orientation. To investigate the internal effects, further analysis was conducted along the vertical direction (b2). Here, the buckypapers $b_{1}$ and $b_{2}$ appear a lighter gray shade than the polymer matrix and carbon fibers due to their lower electron density, leading to increased scattering of the electron beams. Along the cross-section, it becomes apparent that individual prepreg plies in the modified region merge (b3), while the nanotubes within the buckypapers are fully surrounded by the matrix (b4). This suggests that the excess matrix material effectively diffuses into the buckypapers during infiltration. Therefore, the nanotubes (b4) are uniformly and highly concentrated, establishing close contact both with each other and with the carbon fibers. Consequently, a thermal conductivity network forms in this region. Unlike melt processing [24], the carbon nanotube network does not penetrate the prepreg plies, as the nanotubes remain fixed in the buckypaper due to strong attractive forces and mechanical entanglements.

When integrating up to nine buckypapers in the sample $b_{9}p_{1}$ (d1), carbon nanotubes that are no longer fully infiltrated by the polymer matrix become visible on the surface. Macroscopically, this is reflected in the transition from a glossy to a matte-black surface. In the cross-sectional view (d2), along with the first four buckypapers $b_{1}$, $b_{2}$, $b_{3}$, and $b_{4}$, structural defects such as pores, delaminations, or cracks become visible, caused by matrix depletion. These defects (d3) typically contain air, which disrupts the contact between the carbon fibers and the polymer matrix, weakening the fiber–matrix bonds and thereby compromising the mechanical properties.

However, the infiltration process can be deliberately influenced by the periodic arrangement of the prepreg plies. In the sample $b_{5}p_{1}$ (c1), where five buckypapers are each separated by a single ply, the entire surface is incompletely covered with the matrix. With the same number of papers but with two intermediate plies in $b_{5}p_{2}$ (c2), the surface is partially infiltrated, and with four plies in $b_{5}p_{4}$ (c3), it is completely covered. Thus, the more prepreg plies are placed between the buckypapers, the fewer structural defects arise, as more of the polymer matrix becomes available for infiltration.

The prepreg plies of the 8552/IM7 system are basically designed to achieve a high-fiber and a low-matrix-volume content of approximately 42% [15], ensuring excellent specific strengths. Consequently, only a limited amount of matrix material is available for infiltrating the buckypapers, which restricts the number of papers that can be integrated into the material. To determine this manufacturing limit, the samples were analyzed using micro-computed tomography. Figure 4 illustrates, on the right side, a *yz*-tomogram of the $b_{5}p_{1}$ sample along its cross-section.

The five buckypapers are clearly discernible as lighter gray regions compared to the surrounding prepreg plies. This difference arises from the higher density of carbon nanotubes, which results in greater X-ray absorption and thus lower transmittance. Conversely, structural microdefects, prominently visible in the transition zone within the *xy*-tomogram (a2), are represented in a darker gray tone. These matrix-depletion-induced defects typically contain air inclusions with low density, leading to reduced X-ray absorption. This contrast enables a clear distinction between the composite material, the brighter nanotubes, and the darker defect regions.

To quantify the proportion of individual regions within the composite material, a grayscale analysis can be performed, as exemplified in Figure 4 on the left. This involves counting the number of pixels and analyzing their corresponding color values. In the RGB color space, grayscale values range linearly from black (red = 0, green = 0, blue = 0) to white (255, 255, 255), allowing the three color channels to be combined into a single value, *c*. Subsequently, the different regions can be differentiated by introducing two color threshold values: pixels with grayscale values *c* ≤ 108 are assigned to structural defects, while values *c* ≥ 160 correspond to the carbon nanotubes in the buckypapers. Pixels with color values between these thresholds represent the composite material. Consequently, the prepreg plies (a3) consist exclusively of pixels within this middle range. Microdefects in the transition areas (a2) result in a few pixels with values below the lower threshold, while the presence of nanotubes in the buckypapers (a1) leads to pixels with values above the upper threshold. To quantify the defect area *A*_d_ and the nanotube area *A*_n_, the respective areas are normalized by the total composite area *A*_0_.

Figure 5 shows the quantified defect areas on the left axis and the carbon nanotube areas on the right axis as a function of ply depth. In the unmodified material $b_{0}p_{0}$ (a1) neither nanotubes nor defects are detectable over the cross-section. In alternating series, the $b_{2}p_{1}$ laminate (a2) reveals two nanotube peaks near the front surface, corresponding to the two buckypapers. Similarly, the $b_{9}p_{1}$ sample (a3) exhibits nine peaks, as expected. However, the modified region near the buckypaper is overlaid by structural microdefects amounting to a total of *A*_d_/*A*_0_ = 0.7%, which, as discussed in detail, can be attributed to the insufficient amount of matrix available during the infiltration process.

To reduce the defect fraction, additional prepreg plies can be periodically inserted between the buckypapers to increase the available polymer matrix for infiltration. For instance, with five buckypapers and one prepreg ply between each, the defect fraction, as shown in Figure 5(b1), is approximately 0.04%, while it decreases to 0.03% with two prepreg plies (b2) and to barely detectable 0.01% with four prepreg plies (b3). Nevertheless, minor defects remain in these configurations, indicating that the matrix capacity in the 16-ply material is fully utilized during the infiltration of five buckypapers. Although adding more buckypapers causes an accumulation of defects due to matrix depletion, it simultaneously increases the fraction of carbon nanotubes.

This dual effect underscores the challenge of balancing the number of buckypapers with the available matrix, aiming to reduce defects while increasing the nanotube fraction. For a deeper analysis, Figure 6a shows the mass fraction of carbon nanotubes in both the cured laminate and polymer matrix as a function of the number of buckypapers. It is evident that the nanotube proportion increases approximately linearly with the number of buckypapers. Specifically, for each buckypaper, the fraction increases by 0.5 wt.% with respect to the laminate and by 1.6 wt.% relative to the matrix. With a total of five buckypapers, a maximum fraction of 8 wt.% can be incorporated into the matrix without significantly negatively affecting the structural properties. Compared to melt processing [24], up to four times higher proportions can thus be precisely integrated into the material.

However, the integration of buckypapers also affects the laminate’s cross-sectional structure, as illustrated in Figure 6b. For example, the unmodified material has a thickness of approximately 2.02 ± 0.02 mm. For each additional buckypaper, the cross-section increases approximately linearly by 20 μm. This change is relatively modest, as the unprocessed buckypaper, originally 130 μm thick [53], experiences substantial compression during the autoclave curing process. Accordingly, the incorporation of five buckypapers results in an increase of around 100 μm, representing only 5% of the total thickness of the unmodified laminate. This modification thus only marginally impacts the cross-sectional area, which is crucial for preserving the laminate’s mechanical integrity.

In comparison to the previously investigated alternating series of buckypapers within the material, a periodic or symmetric arrangement similarly affects the mass fraction of the nanotubes and the laminate thickness. Consequently, these material properties are primarily determined by the number of buckypapers, while their arrangement plays a crucial role in the proportion of structural defects and, therefore, the material quality, as observed in Figure 5.

Overall, the systems with five buckypapers offer a well-balanced combination of a high fraction of carbon nanotubes and a nearly irrelevant proportion of structural defects within the material volume. It is therefore expected that these modifications provide the greatest potential for enhancing thermal properties while maintaining mechanical integrity.

### 4.3. Material Performance

To validate these expectations, the pathway of heat conduction within the network of carbon nanotubes in the matrix-infiltrated buckypapers must first be examined. This can be indirectly assessed through charge transport, as it follows the path of least resistance. Figure 7 therefore illustrates the normalized electrical conductivity both on the surfaces and in the volume. It is evident that the unmodified material (a1) exhibits a surface conductivity of only 10^−12^ S, primarily due to the insulating effect of the pronounced matrix enrichment. The addition of a single buckypaper on the front side leads to a significant conductivity increase by a factor of 10^8^, as a conductive network is formed in the lateral direction. Further incorporation of additional papers with interleaved prepreg plies brings the conductivity closer to that of untreated buckypaper, at 10^−2^ S, since progressively less polymer matrix is available for infiltration. Simultaneously, the conductivity on the back side increases by a factor of 10^5^, despite the absence of carbon nanotubes. This enhancement can be attributed to the redistribution of excess matrix during the autoclave process, resulting in a locally higher fiber content on the back. In the bulk (a2), the unmodified material demonstrates a specific conductivity of just 10^−11^ S/m. However, the introduction of several buckypapers also only results in a modest improvement by a factor of 10^3^. This suggests that in alternating configurations, the conductive network in the modified region is well-formed, while the non-modified regions remain largely influenced by the surrounding matrix. In contrast, a homogeneous arrangement of buckypapers significantly boosts conductivity by a factor of 10^9^ on the surfaces (b1) and by 10^8^ in the volume (b2). This symmetrical configuration thus drastically enhances electron transport in both lateral and vertical directions, confirming the establishment of a fully integrated conductive network in the material.

To gain a deeper understanding of the transport mechanism, the material is schematically divided into electrical resistances *R* in Figure 7. Modified regions (a1, b1) generally exhibit lower resistances, as enhanced electron transport occurs through the conductive nanotube network in the buckypaper. Therefore, electron transitions between adjacent CNTs can occur via tunneling, allowing electrons to overcome the energy barrier between nanotubes without passing through the insulating matrix [64]. Within the CNTs, electron transport is facilitated by delocalized π-electrons, which move freely along the carbon bonds in the tubular structure [55]. In contrast, regions with enriched electrically insulating matrix materials exhibit higher resistances due to hindered electron transport. Consequently, at the surface, the total resistance is directly influenced by the material’s structural characteristics. Along the cross-section (a2, b2), the total resistance results from the sum of individual resistances connected in series. In an alternating configuration (a2), the total resistance is a combination of low and high individual resistances, as a result of the separation between modified and unmodified regions. This configuration leads to only a minor reduction in overall resistance compared to the original material. In contrast, a symmetrical arrangement (b2) results in a total resistance determined by the repeated individual resistances of uniform sections. These resistances, which lie between the alternating individual resistances, lead to a significantly lower total resistance, indicating the overall better electrical transport properties. Similarly, these regions can be interpreted as thermal resistances, providing a basis for drawing parallels to the thermal behavior of the modified materials.

Building on this perspective, the effects of the integrated buckypapers on the thermal properties of the materials are now examined. For this purpose, Figure 8 illustrates the thermal conductivity (a1, a2) along the sample cross-section. The unmodified sample shows a thermal conductivity of just 0.772 ± 0.010 W/(m·K) at room temperature, emphasizing its insulating behavior, primarily due to the polymer matrix. As expected, the integration of buckypapers improves the thermal conductivity by creating more efficient conductive pathways. For example, the conductivity increases by approximately 13% (a1) with five buckypapers arranged alternately, while it rises by 20% (a2) in a symmetric arrangement. In comparison, a significantly better thermal performance is observed when the buckypapers are uniformly arranged within the laminate, as this configuration establishes a continuous and efficient network for heat transfer.

An even higher thermal conductivity is expected exclusively in the longitudinal direction of the sample, owing to the anisotropic structure of the carbon fibers and nanotubes. In this direction, a highly ordered, graphitic crystal structure with covalent bonds facilitates particularly efficient heat transport along direct pathways. However, unlike electrical conductivity, thermal conductivity does not exhibit a sharp increase. This is because heat transfer occurs not through electrons, but via quantized lattice vibrations of phonons. Since the nanotubes are only weakly bonded through van der Waals forces, a significant thermal contact resistance arises at their interfaces, leading to phonon scattering and thus disrupting the heat conduction within the network [24,29]. As a result, the improvement in heat transport is more modest compared to electron transport.

The specific heat capacity (b1, b2) of the unmodified material is approximately 0.908 ± 0.004 J/(g·K) under ambient conditions. Integrating buckypapers into the composite material increases their specific heat capacity, as the nanotubes contribute to enhanced thermal storage. For instance, the incorporation of five papers results in a similar capacity boost of around 5% for the alternating arrangement (b1) and 7% for the symmetric distribution (b2) over the cross-section. The placement of the buckypapers within the material thus has a negligible effect on the specific heat capacity. Therefore, it is primarily the concentration of carbon nanotubes that drives the moderate improvement in thermal energy storage.

The density of the non-modified material (c1, c2) is 1.621 ± 0.001 g/cm^3^ under standard conditions. The introduction of buckypapers leads to an unexpected decrease in material density. Specifically, with five buckypapers, the density drops by approximately 1.3% for the alternating arrangement (c1) and 0.1% for the symmetric series (c2). This decrease can be attributed to several factors: First, the incorporation of buckypapers increases the total volume more significantly than the total mass, owing to the lightweight nature of the nanotubes. Second, at higher paper numbers, incomplete infiltration of the polymer matrix may result in microdefects, trapping air pockets with a lower density than the material itself. Despite these factors, the reduction in density remains minimal compared to the original material. Consequently, the modification offers a slight improvement in thermal properties without negatively affecting the material’s specific weight.

Finally, the mechanical behavior is examined to assess the impact of the buckypaper modification on the material’s structural integrity. To this end, Figure 9(a1,a2) illustrates the thermo-mechanical properties obtained from dynamic mechanical analysis. During the heating process, the non-modified material transitions from a rigid, glassy state to a soft, viscous state at a glass transition temperature of approximately 230.7 ± 1.0 °C. The incorporation of buckypapers in an alternating arrangement within the material (a1) leads to a slight temperature increase of up to 233.9 ± 1.4 °C. This delayed softening of the material could be attributed to various structural and chemical changes; the carbon nanotubes in the buckypapers may act as nanoscale fillers, restricting the mobility of the polymer molecular chains. Additionally, the nanotubes’ large specific surface area could enhance interactions with the polymer matrix. This, in turn, could increase the crosslinking density within the material, contributing to the higher glass transition temperature. However, since a symmetrical arrangement of the buckypapers (a2) leads to a similar temperature change, the concentration of nanotubes integrated into the material primarily influences a slightly higher resistance to thermal and mechanical stresses.

For a comprehensive analysis, Figure 9(b1,b2) also illustrates the normalized strengths for varying numbers of buckypapers in the composite material. The unmodified sample has an interlaminar shear strength of 71.2 ± 0.1 MPa, a compressive strength of 746 ± 32 MPa, a bending strength of 633 ± 4 MPa, and a tensile strength of 890 ± 19 MPa. The integration of buckypapers leads to a noticeable reduction in strength, indicating that carbon nanotubes do not reinforce the nanocomposite. Upon closer inspection, when up to five papers are alternately arranged (b1), a slight strength reduction of about 10% is observed. This decline can primarily be attributed to the increased cross-sectional area of the nanocomposites, as these samples absorb forces comparable to those of the base material during mechanical loading. Only with a higher number of papers does the strength drop to 63% of its original value. This behavior is linked to the formation of microdefects, caused by the limited matrix availability. As a result, force transfer from the matrix to the carbon fibers is hindered due to weak fiber–matrix bonding. However, the mechanical properties can be significantly influenced by the arrangement of the buckypapers. In a symmetrical arrangement (b2) with five buckypapers, the impact on strength is minimal compared to the alternating series, as these samples contain only a few defects. With nine buckypapers, this effect improves by about 24%. Nevertheless, the strength still only reaches 87% of the original value and is thus no longer within the acceptable range for high-performance materials. Thus, the systems with five buckypapers represent, as expected, the best combination of thermal properties without significantly compromising mechanical integrity.

### 4.4. Simulation of Thermal Properties

The thermal properties determine the ability of heat conduction and storage in advanced materials. To describe this thermal behavior, Bibinger et al. [24] recently published an innovative yet simple model. In this approach, the thermal conductivity of a composite material $\lambda_{\mathrm{mod}}$ is calculated using the rule of mixtures, as outlined in Equation (Equation 1):(1)$$\lambda_{\mathrm{mod}}=\left(1-\omega_{p,l}\right)\lambda_{l}+\omega_{p,l}\cdot \lambda_{p,\mathrm{eff}}$$
where $\omega_{p,l}$ denotes the nanotube mass fraction within the laminate, $\lambda_{l}$ represents the experimentally determined thermal conductivity of the unmodified laminate, and $\lambda_{p,\mathrm{eff}}$ indicates the effective thermal conductivity of the carbon nanotubes embedded in the polymer matrix. Unlike the theoretical assumptions in conventional serial or parallel models [29,65], this approach focuses on the actual heat conduction behavior of the nanoparticles when integrated as a filler in composite materials. To this end, both the regions with lower thermal resistances within the tubular structure and those with higher resistances between adjacent nanotubes are considered. These regions together form the total thermal resistance, which can be indirectly captured through its inverse relationship with the effective thermal conductivity. To determine the effective thermal conductivity, Figure A1 in the Appendix A compares the calculated to the experimentally measured thermal conductivity at various temperatures. The calculation is performed using Equation (Equation 1) by varying the conductivity of the nanotubes for a given particle fraction and the conductivity of the laminate. The effective conductivity of the nanoparticles is found when the best agreement between the calculated and experimental values is obtained. To meet this condition, the method of least squares is applied by maximizing the coefficient of determination. For example, in the alternating arrangement of the buckypapers (a1), the highest coefficient of determination of $R^{2}$ = 0.937 is achieved at an effective thermal conductivity of 4.95 W/(m·K) for the integrated nanotubes, whereas in the symmetric arrangement (a2), $R^{2}$ = 0.981 is obtained for 6.92 W/(m·K). Consequently, these nanotubes attain an effective thermal conductivity that is 6 to 9 times higher than that of the composite material itself. However, as observed in the melting process [24], these calibrated effective conductivities are significantly lower than the theoretical ranges, which can reach 2000 to 6000 W/(m·K) for CNTs [66]. This discrepancy arises because the nanotubes cannot fully exploit their theoretical potential within the composite, due to factors such as insufficient nanoparticle network density, misalignment of the nanotubes, defects and impurities in the nanotubes, and significant thermal resistance between adjacent particles, all of which limit the effective thermal conductivity.

The effective specific heat capacity and density can be determined analogously using Equation (Equation 1), as illustrated in Figure A1. A summary of the experimentally determined values for the unmodified material and the calibrated effective values for the nanotubes is provided in Table 1. The nanotubes exhibit effective specific heat capacities of approximately 1.77 ± 0.02 J/(g·K) and an effective density of about 1.27 ± 0.16 g/cm^3^, regardless of the arrangement of the buckypapers within the material. These effective values deviate significantly from the theoretical values for multi-walled carbon nanotubes, which have a specific heat capacity of approximately 0.8 J/(g·K) [67] and a density of 1.6 g/cm^3^ [67]. This difference results from the fact that the effective values capture the behavior of the nanotubes as part of the matrix and not their properties in the free state. In the case of thermosetting plastics like epoxy resins, the matrix typically exhibits specific heat capacities between 1.7 and 2.0 J/(g·K) [68] as well as a density of approximately 1.3 g/cm^3^ [15]. Consequently, the interaction between the nanotubes and the matrix causes the effective properties to align more closely with those of the plastic. Overall, the nanoparticles nearly double the specific heat capacity while maintaining only one-third of the density of the unmodified composite material. As a result, the embedded nanoparticles significantly enhance both the thermal storage and the lightweight potential of the composite material.

To validate the calibrated effective values, Figure 8 compares the modeled data with the experimental results, including a 5% margin of error. This comparison highlights the strong correlation between the modeled and experimental values. For instance, when five buckypapers are incorporated into the composite material in an alternating arrangement (a1), the thermal conductivity is calculated as $\lambda_{\mathrm{mod}}$ = (1 − 0.027)·0.772 W/(m·K) + 0.027·4.95 W/(m·K) = 0.885 W/(m·K). This value aligns well with the experimentally determined thermal conductivity of 0.871 ± 0.032 W/(m·K). Similarly, when the same number of buckypapers are arranged symmetrically (a2), the higher effective thermal conductivity of the nanoparticles results in a calculated value of 0.938 W/(m·K), which closely matches the experimental result of 0.920 ± 0.005 W/(m·K). Beyond thermal conductivity (a1, a2), the approach also accurately predicts the specific heat capacity (b1, b2) and density (c1, c2) of the nanocomposite. Consequently, these findings underscore the reliability of the model in describing the thermal behavior of modified composite materials with buckypapers.

Under thermal loading, these properties determine the heat conduction in the composite material. Heat conduction is naturally described by the following unsteady-state heat conduction equation [69]: (2)$$\frac{\partial T(z,t)}{\partial t}=a\cdot \frac{\partial^{2}T\left(z,t\right)}{\partial z^{2}}\;\;\mathrm{with}\;\;a=\frac{\lambda }{\rho \cdot c_{p}}$$
where *T* denotes temperature, *z* represents the material depth along the cross-section, *t* depicts the exposure time, and *a* illustrates the thermal diffusivity. Therefore, thermal diffusivity characterizes the rate at which a temperature change propagates through a material. According to (Equation 2), it is defined as the ratio of a material’s ability to conduct heat to its capacity for storing heat. A high thermal conductivity λ facilitates efficient heat transfer through the material, enabling rapid heat dissipation and distribution. Conversely, a high volumetric heat capacity $\rho \cdot c_{p}$ allows the material to absorb more heat energy before undergoing a significant temperature change, promoting effective heat storage and temperature stabilization.

As shown in Table 1, the base material exhibits a thermal diffusivity of 0.525 ± 0.007 mm^2^/s, resulting from its relatively low thermal conductivity of 0.772 W/(m·K) compared to its volumetric heat capacity of 1.47 J/(cm^3^·K). This balance suggests that the material primarily functions as a thermal storage medium. The integration of carbon nanotubes significantly enhances the thermal performance. When arranged in an alternating buckypaper configuration, the nanotubes achieve a five-fold increase in thermal diffusivity of 2.55 ± 0.26 mm^2^/s. This improvement is attributed to a substantial increase in effective thermal conductivity to 4.95 W/(m·K), while the effective volumetric heat capacity remains relatively moderate at 1.94 J/(cm^3^·K). As a result, the nanotube’s structure behaves more like a heat sink. For example, integrating five buckypapers into the $b_{5}p_{1}$ sample increases the thermal diffusivity to approximately *a* = (1 − 0.027)·0.525 mm^2^/s + 0.027·2.55 mm^2^/s = 0.580 mm^2^/s. In a symmetric buckypaper arrangement, nanotubes even achieve a slightly higher diffusivity of 2.73 ± 0.27 mm^2^/s. However, due to its higher volumetric heat capacity, the resulting effective thermal diffusivity of the $b_{5}p_{4}$ specimen remains comparable at 0.585 mm^2^/s. Thus, the thermal diffusivities for different buckypaper configurations are nearly identical, indicating that heat distribution is predominantly governed by the concentration of carbon nanotubes rather than their specific arrangement.

### 4.5. Thermo-Induced Damage

The effects of enhanced heat conduction in nanocomposites on their resistance to one-sided thermal stress are now examined. The focus is on the modified sample, consisting of five buckypapers interleaved with single prepreg plies, as it represents a well-balanced combination of high nanotube concentration and improved thermal properties. For this purpose, Figure 10 illustrates the temperature profile *T*(*z, t*) for both the unmodified $b_{0}p_{0}$ sample (a1) and the modified $b_{5}p_{1}$ specimen (b1) under a heat flux of 50 kW/m^2^. During thermal loading, the impinging radiative heat is primarily absorbed on the front side and conducted to the cooler back side, forming a temperature gradient along the cross-section. In this process, the unmodified material (a1) initially heats up at a rate of approximately 19 °C/s. With increasing loading time, the heating rate slows down according to the lower temperature rise. This effect is partly caused by the increased reflection of incoming radiation at higher temperatures, as described by the Stefan–Boltzmann law, where radiated power increases with the fourth power of temperature [70]. As a result, after, for example, 20 s, the material reaches an average temperature of 327 ± 14 °C. In contrast, the modified material (b1) heats up at 14 °C/s, reaching only 252 ± 20 °C under the same conditions. Heat conduction is thus more efficient in the modified material, resulting in lower energy input. However, prolonged exposure also intensifies the temperature gradient along the ply depth.

For a more detailed analysis, Figure 10 presents the temperature difference between the front and back sides of the non-modified (a2) and modified (b2) samples. After a brief warm-up phase, both materials stabilize at a temperature difference of around 25 °C for several seconds. With continued exposure, the temperature difference in the original material (a2) gradually increases to 41 °C, while the modified material (b2) quickly rises to nearly 100 °C during the loading phase. This suggests a more uneven heat distribution and steeper temperature gradient in the nanocomposite (b2), despite its enhanced thermal conductivity. To identify the cause, the initiation times of various thermo-induced damage processes are also shown in this figure. In the base material (a2), matrix degradation begins from around 8 s, with negligible effects on temperature development. A slight increase can only be observed after 15 s, attributed to the formation of structural damage such as delamination, impairing heat conduction by trapping insulating pyrolysis gases. Beyond 21 s, the temperature gradient rises sharply as the polymer matrix gradually decomposes into these gases and solid pyrolysis residues, further hindering heat transfer. In comparison, thermal damage in the modified material occurs later, but its impact on temperature is more pronounced. This indicates that temperature development is primarily determined by these damage processes, warranting a closer examination in the following section.

First, the matrix degradation is analyzed using infrared spectroscopy. Figure 11a shows the infrared spectra from the front side of the materials with alternating buckypaper arrangements. The band at 1510 cm^−1^ is assigned to the ν(C=C) ring vibration in the aromatic structure of the epoxy resin (EP), while the band at 1486 cm^−1^ corresponds to analogous vibrations in polyethersulfone (PES) [71,72]. To quantify matrix degradation, band intensities are determined through baseline-corrected integrations within a defined wavenumber range, and the ratio $I_{8552,d}$ = $I_{1510\;{\mathrm{cm}}^{-1}}$/$I_{1486\;{\mathrm{cm}}^{-1}}$ is calculated. In the unmodified sample $b_{0}p_{0}$, this ratio is approximately 1.09, whereas in the nanocomposites, the intensity diminishes progressively with rising buckypaper number, until no infrared signal (ns) is detectable. This development arises from various absorption and scattering effects. During measurement, infrared rays penetrate a few micrometers into the material and excite molecular vibrations. However, energy-dispersive X-ray spectroscopy analyses reveal that the carbon content on the surface increases from 65 to 88% due to the modifications. This increase leads to two effects: First, more incoming infrared radiation is absorbed by the modified surface, reducing the signal-to-noise ratio and thus the accuracy of the infrared band intensity determination. Second, Rayleigh scattering occurs at the infiltrated paper surface, as the structures of carbon nanotubes are significantly smaller than the infrared wavelength [73]. This wavelength-dependent scattering deflects shorter wavelengths and thus higher wavenumbers more strongly than longer wavelengths [73]. Consequently, as the carbon content rises, the baseline drops more steeply, distorting the band intensity measurements. Nevertheless, to compare thermo-induced matrix degradation between the original and modified materials, the infrared intensities are considered exclusively in the prepreg regions.

For this, Figure 11 shows the quantified matrix degradation based on infrared intensity along the ply depth. In the original, untreated material (b1), the intensity remains constant at approximately 1.03 ± 0.03 over the entire cross-section. Only under one-sided thermal stress does the intensity decrease towards the front side after about 20 s due to the decomposition of the epoxy resin. In this initial stage, secondary alcohols in the polymer are predominantly dehydrated [74]. As thermal stress increases, the intensity loss continues along the ply depth. Therefore, the allylic amine groups in the polymer can be cleaved and converted into volatile components or solid char residues [75,76]. After around 30 s, no infrared signal is detectable on the front side, indicating, among other degradation processes, thermoplastic decomposition. The dominant mechanism involves the carbonization of polyethersulfone, releasing sulfur dioxide from the sulfonic group and phenol from the ether group [77,78]. Shortly thereafter, both the thermosetting and thermoplastic components are decomposed throughout the material. In comparison, the modified material (b2) shows only a moderate and steady intensity decrease up to 30 s. The damage within the volume of the nanocomposite is thus significantly less pronounced. However, due to the incomplete matrix infiltration of the buckypaper on the front side, the bonding to the adjacent prepreg ply is slightly weakened, so that even moderate matrix degradation can negatively affect both its adhesive and structural properties.

Structural damage was assessed using micro-computed tomography. Figure 12 illustrates the damaged area as a function of ply depth. In the course of the heating process, delaminations become detectable in the base material (a1) after approximately 20 s. This damage phenomenon results from a combination of thermal stresses within the material and the decomposition of the polymer matrix. Under thermal loading, internal stresses arise due to the mismatch in thermal expansion properties: carbon fibers contract along their axial direction, while the polymer matrix expands isotropically. In the glass transition region, the epoxy resin additionally begins to decompose. Once a critical stress threshold is reached, accompanied by significant matrix degradation, thermo-induced structural damage occurs during relaxation. As heating continues, the damaged area grows due to delamination propagation. Therefore, local maxima can appear along the cross-section, resulting from a stronger detachment of individual plies (a2). However, structural damage in the sample with five integrated buckypapers (b1) forms slightly later. Accordingly, after, e.g., 25 s, the damage in the modified material is less pronounced, both in the quantified analysis (a1 → b1) and in the qualitative representation of the *xz*-tomograms (a2 → b2). Under the same loading conditions, the nanocomposite thus consistently exhibits less structural damage. At longer exposure times, however, more pronounced damage gradients develop, rapidly decreasing toward the back side. The cause of this damage distribution will be further examined below.

To this end, Figure 13a shows the mass fraction of the materials after thermal exposure. Initially, the samples gain weight, indicating the formation of oxidation products near the surface. Only after about 20 s does the original $b_{0}p_{0}$ sample rapidly lose mass, while the modified materials, especially the $b_{5}p_{1}$ specimen, show a delayed decrease. This mass depletion results from the decomposition of the polymer matrix into volatile pyrolysis gases, which diffuse to the front side and are released to the environment. This process also affects the interply adhesion, which can be indirectly characterized by the development of the cross-section shown in Figure 13(b1). Similarly to the weight loss, the material thickness begins to change after approximately 20 s, and subsequently increases by up to 37 ± 6% for the $b_{0}p_{0}$ sample, while the modified $b_{2}p_{1}$ specimen even reaches 64 ± 6%. This expansion occurs because decomposition products are not only released on the surface but also trapped between the plies. In the unmodified material (b2), less expansion is observed, since the pyrolysis gases can escape more freely, as shown in the surface image, where the depleted matrix exposes the carbon fibers in the 45° direction. In contrast, the modified material (b3) expands more noticeably due to the infiltrated buckypaper, which inhibits the diffusion of volatile products. Its dense carbon nanotube network acts as a physical barrier, causing pyrolysis gases to accumulate and inflate the buckypaper. As a result, this region provides thermal insulation, reducing heat flux and protecting the underlying material. This ultimately explains the sharp gradient drop in heat input (Figure 10), matrix degradation (Figure 11), and structural damage (Figure 12). However, in the $b_{5}p_{1}$ sample, the cross-sectional expansion is even less pronounced than in the original material. This is due to delayed damage processes from the improved thermal properties and the incomplete infiltration of the buckypaper, which allows the gases to escape more easily.

### 4.6. Thermal Durability

The impact of thermal loading on the mechanical material properties is explored below. For this, Figure 14 illustrates the interlaminar shear strength of samples with buckypaper in different arrangements, determined by the short beam test. This test was selected because the material reacts highly sensitive to even minimal thermal damage under shear loading, such as slight weakening of fiber–matrix bonding [16,17], enabling the assessment of early-stage damage on the mechanical performance of CFRP. The original material (a1) exhibits an interlaminar shear strength of 71.2 ± 0.1 MPa. With increasing thermal loading, strength gradually decreases due to minor damages like surface oxidation of the polymer matrix, microcracks at the fiber–matrix interface, or initial matrix degradation. After about 15 s, the strength drops sharply with the formation of structural damage, as the transfer of shear forces from the polymer matrix to the reinforcing fibers is impaired. Subsequently, the strength continues to decline until almost no shear forces can be absorbed, because the depleted matrix and accumulated pyrolysis gases within the material ultimately result in the loss of interlaminar adhesion. In comparison, the nanocomposites with alternating (a1) and periodic (b1) buckypaper arrangements experience a slight delay in strength reduction. Therefore, the $b_{9}p_{1}$ sample is notable for its delayed strength loss under thermal loading, despite its lower initial strength of 52.0 ± 1.2 MPa.

To quantify this improvement, the irradiation times of the strength values for the modified materials were multiplied by varying delay factors until the best match with the reference curve of the non-modified material was achieved using the least squares method. For further analysis, Figure A2 in the Appendix A presents the transformed reference curves alongside the calculated delay factors, while Figure 14 directly shows these factors, reflecting the retardation of strength loss due to the enhanced thermal properties of the incorporated carbon nanotubes. It is obvious that with increasing incorporation of buckypapers (a2, b2), the loss of strength initially begins later. The maximum improvement occurs with five buckypapers, yielding a 17% delay in the alternating arrangement (a2) and 20% in the periodic arrangement (b2). Both configurations thus show similar improvement potentials. However, at even higher paper numbers, the enhancement diminishes due to incomplete infiltration, as the defects induced by an insufficient matrix weaken the thermal stability. Consequently, the paper’s arrangement is crucial for the infiltration process and material properties, while the concentration of carbon nanotubes primarily determines the potential for enhancing thermal endurance. Overall, the systems with five buckypapers thus represent the most advanced materials with the highest heat resistance.

## 5. Conclusions

This study aimed to enhance the thermal stability of polymer-based composites while preserving their specific mechanical properties. To achieve this, sixteen prepreg plies were combined with buckypapers in varying quantities and positions within the laminate. In the alternating arrangement, the papers were concentrated near the front region, whereas in the periodic series, they were progressively shifted deeper, ultimately achieving an even distribution in the symmetric configuration. Subsequently, the nanocomposites were examined for their manufacturing process, properties, and thermal stability.

The results indicate that the prepreg plies provide excess matrix for effective buckypaper infiltration. As a result, up to five papers can be integrated into the composite without inducing defects. In this modification, a single buckypaper on the surface is sufficient to form a continuous lateral carbon nanotube network, while a symmetrical arrangement of multiple buckypapers is necessary to establish a pronounced network through the thickness. Overall, both thermal conductivity and heat storage improve noticeably without compromising weight or mechanical strength. However, compared to theoretical expectations for loose nanotubes, the improvements are modest due to significant thermal resistance between the matrix-embedded nanotubes. Nevertheless, the modified material heats up more slowly than the original, owing to its more efficient heat conduction, which substantially delays thermo-induced damage and strength loss. Whether heat is dissipated more efficiently on the surface with an alternating arrangement or uniformly through a symmetrical arrangement, the impact on mechanical properties remains negligible. Ultimately, the nanotube fraction is the key factor for the material’s thermal resistance.

Compared to the melt process, integrating buckypapers enables a simpler and faster incorporation of up to four times higher nanotube content of 8 wt.% without requiring dispersion. Consequently, thermal stability can be improved by up to threefold. However, these performance gains come at a cost: high-quality carbon nanotubes are already expensive, and the additional processing steps required to fabricate buckypapers further increase overall material costs. Therefore, the improvements in thermal stability must be carefully weighed against economic feasibility, especially for large-scale applications.

The improvement potential is currently limited by the low excess matrix content in the prepreg plies. To boost performance, additional buckypapers combined with pure polymer matrix layers could be inserted between the prepreg plies. This would enable higher nanotube concentrations without the risk of matrix depletion defects. It is crucial to note, however, that the additional polymer matrix should be kept as low as possible to maintain the lightweight benefits.

Additionally, the thermal performance of the nanocomposites could be further enhanced by incorporating buckypapers in a three-dimensional configuration or by strategically positioning them along the edges of the laminate. This strategy aims to significantly improve thermal conductivity in both the lateral and vertical directions. As a result, heat could be dissipated even more efficiently under thermal loading.

In the aerospace industry, buckypapers can now also be used as safeguards against various thermal conditions. However, a serious drawback of composite materials is that thermo-induced damage, such as delaminations, often goes undetected to the naked eye, posing a significant safety risk. Fortunately, with buckypaper-modified materials, this issue could be resolved. During heating, pyrolysis gases are trapped on the surface, causing premature bulging. This visual change serves as an early warning system for thermal damage, further improving safety.

## Figures and Tables

**Figure 1 nanomaterials-15-01081-f001:**
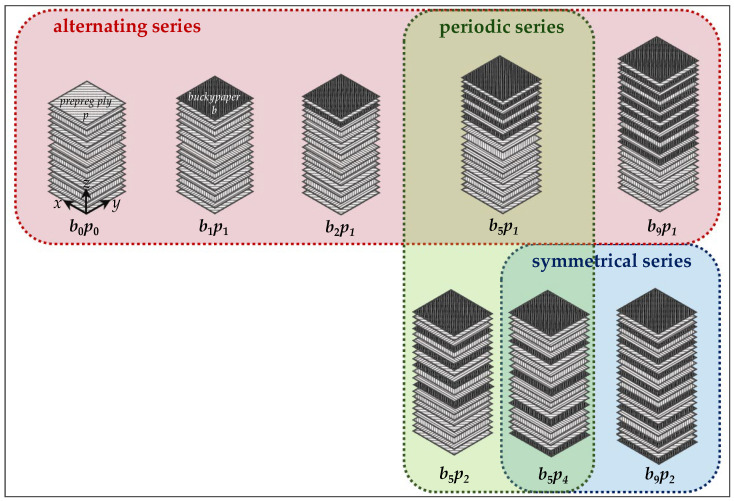
Schematic fabrication process of nanocomposites with alternating, periodic, and symmetrical series of buckypapers.

**Figure 2 nanomaterials-15-01081-f002:**
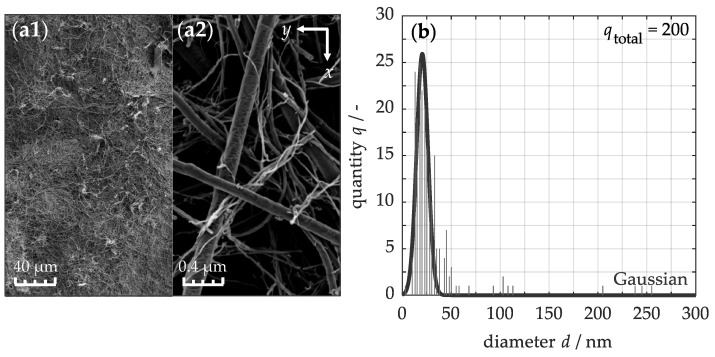
SEM micrographs of a buckypaper at lower (**a1**) and higher magnification (**a2**), including the statistical particle size distribution of the carbon nanotube cross-sections (**b**).

**Figure 3 nanomaterials-15-01081-f003:**
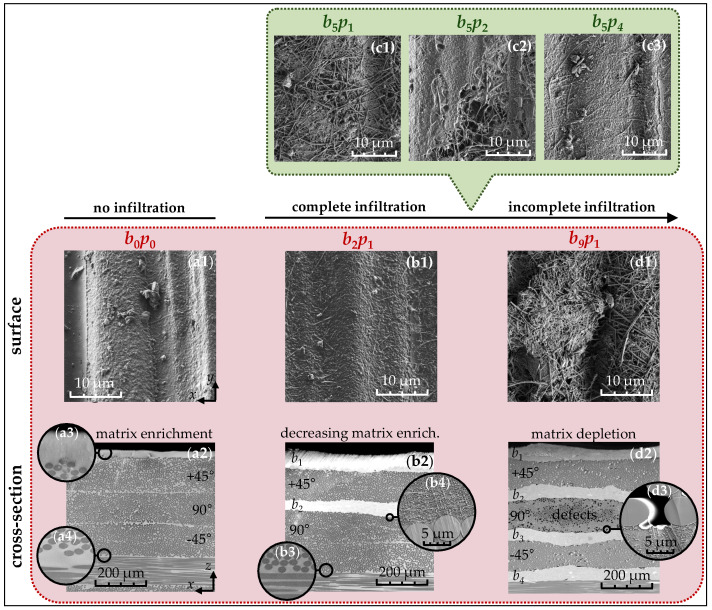
Visualization of the infiltration process at the surface and along the cross-section of the samples $b_{0}p_{0}$ (**a1**–**a4**), $b_{2}p_{1}$ (**b1**–**b4**), $b_{5}p_{1,2,4}$ (**c1**–**c3**), and $b_{9}p_{1}$ (**d1**–**d3**) based on SEM micrographs.

**Figure 4 nanomaterials-15-01081-f004:**
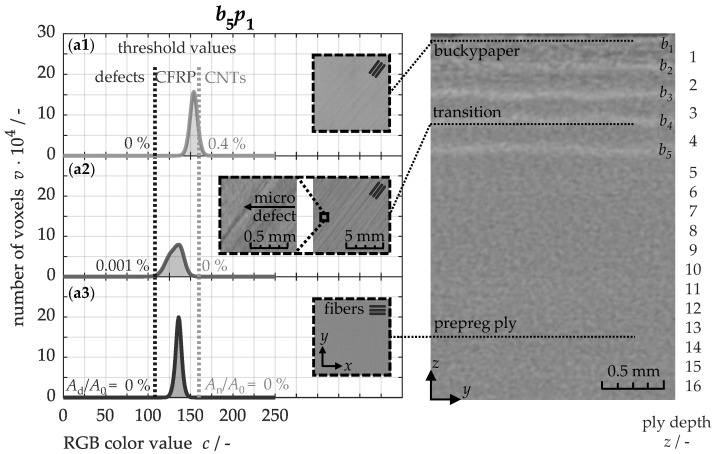
Exemplary gray value analysis of micro-computed tomograms (**a1**–**a3**) for quantifying defect and nanotube areas within the modified $b_{5}p_{1}$ material.

**Figure 5 nanomaterials-15-01081-f005:**
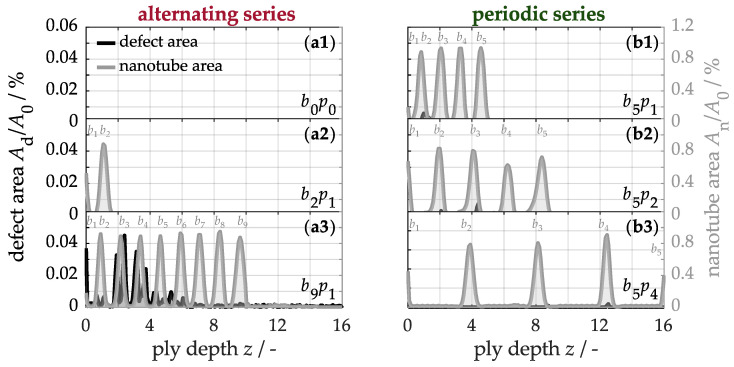
Quantification of defect and nanotube areas along the ply depth of samples with buckypapers in alternating series (**a1**–**a3**) and periodic series (**b1**–**b3**).

**Figure 6 nanomaterials-15-01081-f006:**
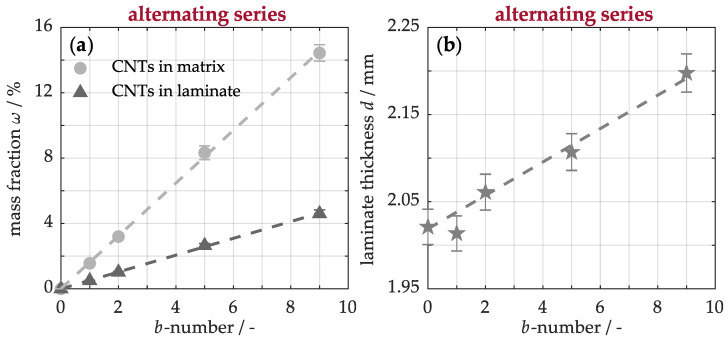
Effect of the number of incorporated buckypapers on carbon nanotube mass fraction (**a**) and laminate thickness (**b**) in the modified composites.

**Figure 7 nanomaterials-15-01081-f007:**
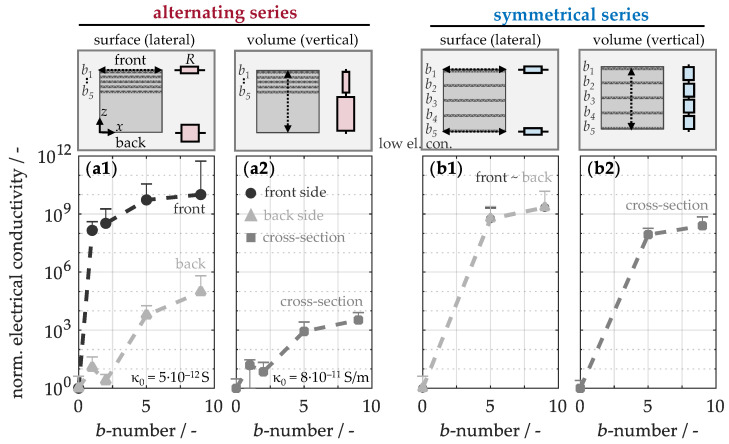
Normalized electrical conductivity on the surface and in the volume of composites with buckypaper in alternating (**a1**,**a2**) and symmetrical arrangements (**b1**,**b2**), including a schematic representation of the electrical resistances within the material.

**Figure 8 nanomaterials-15-01081-f008:**
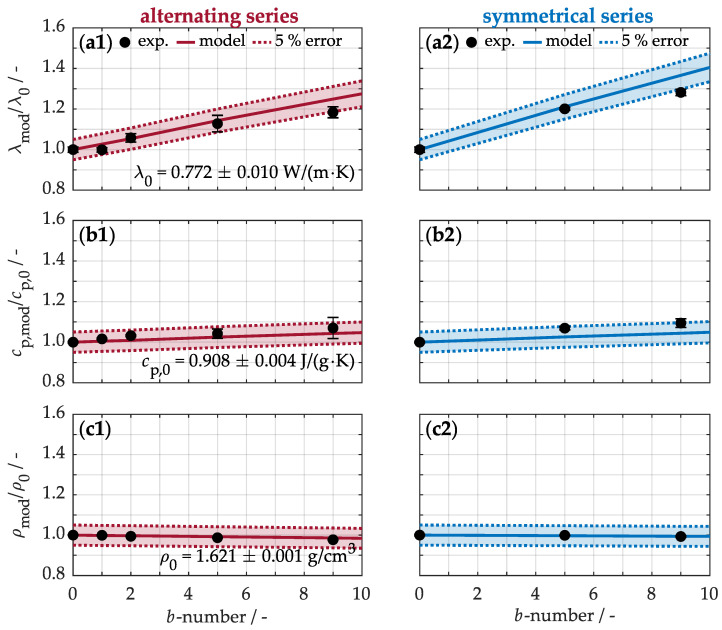
Experimental and calculated normalized thermal conductivity (**a1**,**a2**), specific heat capacity (**b1**,**b2**), and density (**c1**,**c2**) as a function of buckypaper number. The model calibration is presented in Figure A1.

**Figure 9 nanomaterials-15-01081-f009:**
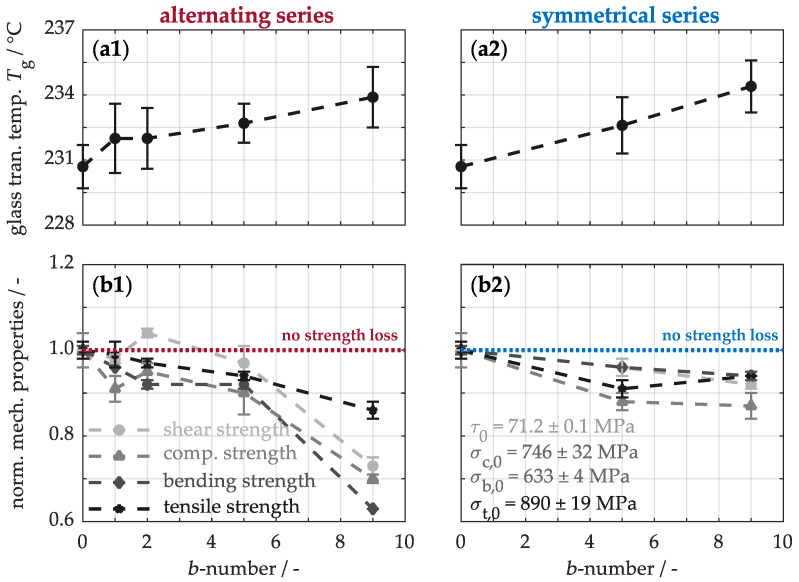
Mechanical behavior of the modified composites with respect to their glass transition temperatures (**a1**,**a2**) and strengths (**b1**,**b2**).

**Figure 10 nanomaterials-15-01081-f010:**
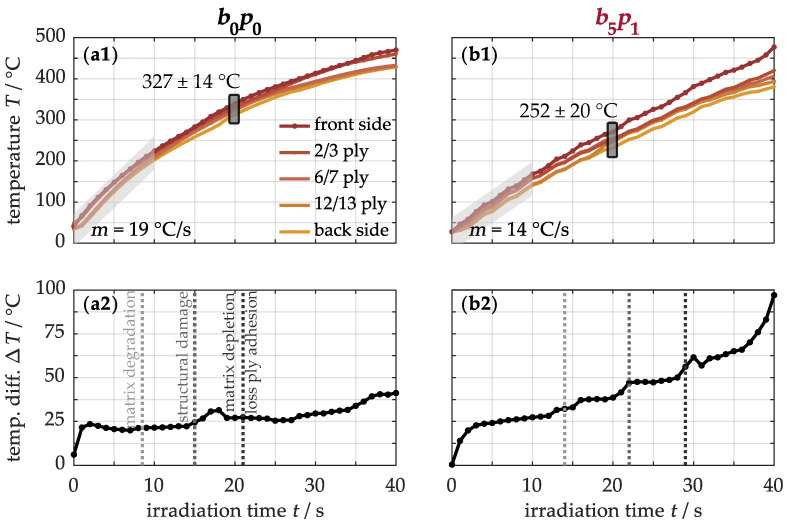
Temperature profiles *T*(*z, t*) (**a1**,**b1**) recorded with five attached Type K thermocouples at different ply depths, along with temperature differences between the front and back sides (**a2**,**b2**) for non-modified and modified materials under one-sided thermal loading with a heat flux of 50 kW/m^2^.

**Figure 11 nanomaterials-15-01081-f011:**
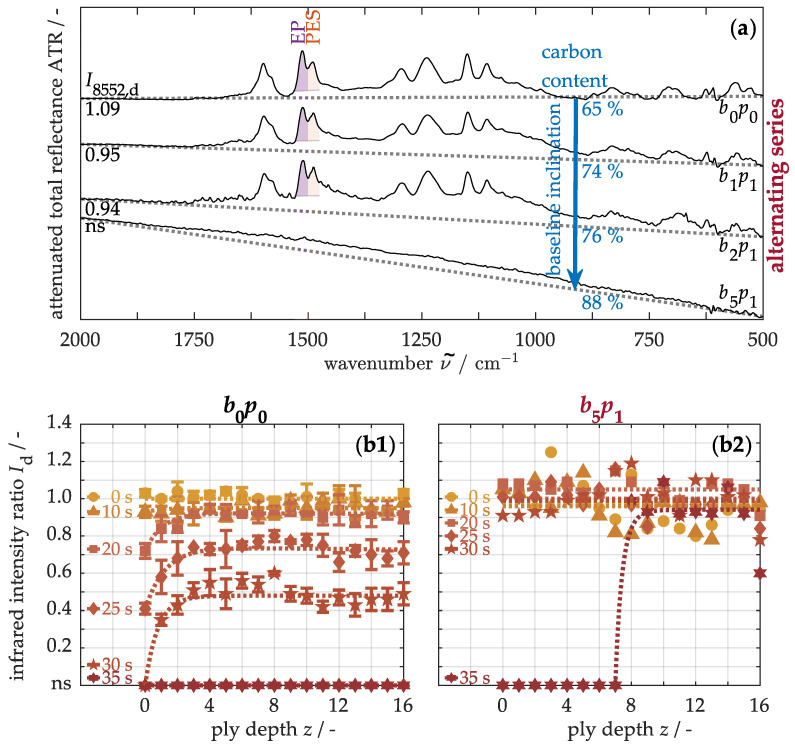
Representative infrared spectra from the front side of materials with alternating arrangements of buckypapers, including carbon content determined by EDX (**a**), and quantified matrix degradation along the cross-section of the unmodified $b_{0}p_{0}$ (**b1**) and modified $b_{5}p_{1}$ samples (**b2**) using infrared spectroscopy.

**Figure 12 nanomaterials-15-01081-f012:**
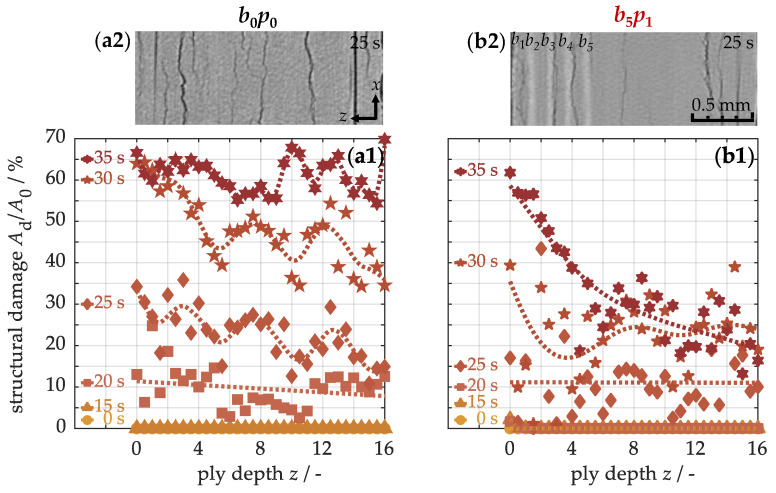
Quantification of structural damage along the ply depth of the unmodified $b_{0}p_{0}$ (**a1**) and modified $b_{5}p_{1}$ specimen (**b1**) using micro-computed tomography, including visualization of exemplary tomograms (**a2**,**b2**).

**Figure 13 nanomaterials-15-01081-f013:**
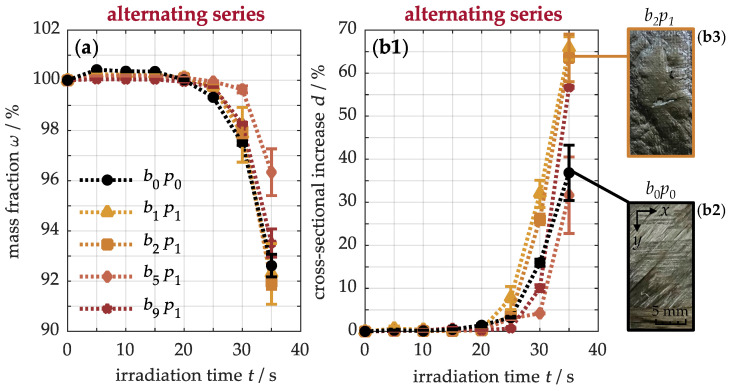
Evaluation of the thermo-induced change in mass (**a**) and thickness (**b1**) of materials with alternating arrangements of buckypapers. The physical barrier of the buckypaper is exemplified through photographs (**b2**,**b3**) of the sample surface.

**Figure 14 nanomaterials-15-01081-f014:**
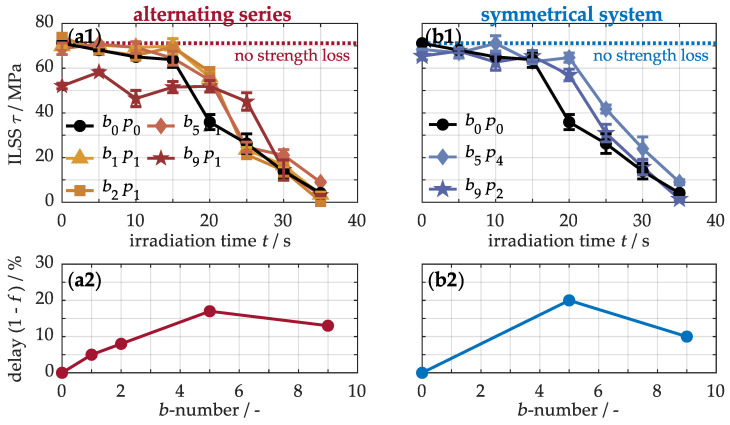
Comparison of interlaminar shear strength (**a1**,**b1**) and percentage delay in strength loss between non-modified and modified samples (**a2**,**b2**). The transformation is shown in Figure A2.

**Table 1 nanomaterials-15-01081-t001:** Comparison of the effective thermal properties of the integrated nanotubes in the modified samples with alternating and symmetrical buckypaper arrangements to the unmodified material. The model calibration is shown in Figure A1.

Effective Thermal Properties	CFRP	Alternating	Symmetrical
Thermal Conductivity/W/(m·K)	0.772 ± 0.010	4.95 ± 0.50	6.92 ± 0.69
Specific Heat Capacity/J/(g·K)	0.908 ± 0.004	1.75 ± 0.18	1.78 ± 0.18
Density/g/cm^3^	1.621 ± 0.001	1.11 ± 0.11	1.42 ± 0.14
Thermal Diffusivity/mm^2^/s	0.525 ± 0.007	2.55 ± 0.26	2.73 ± 0.27

## Data Availability

Data are contained within the article.

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
