# Peer review of "Buckypapers in Polymer-Based Nanocomposites: A Pathway to Superior Thermal Stability"

_nanomaterials, 2025, doi:10.3390/nano15141081_

Round 1
Reviewer 1 Report
Comments and Suggestions for Authors
This manuscript deals with thermal properties of buckypaper composites. The materials characterizations is quite extensive, there are two main designs tested, the influence of various parameters on thermal properties is reported.
My main comment is that buckypapers are expected to exhibit highly anisotropic thermal conductivity it this does not seem to be addressed too much. In fact, in-plane thermal conductivity can easily be 1 order of magnitude higher than through-plane thermal conductivity. Although the authors focus on through-plane thermal conductivity, it would be interesting to test in-plane or lateral thermal conductivity as well. Some 3D design, or even just something simple with “buckypaper vertical walls” at the edge of the materials could drastically increase “in plane” (i.e. vertical) thermal conductivity.
The introduction could be improved citing the various parameters (CNTs diameter, length, number of layers (or layer thickness), layer compression...) that have huge influence on buckypaper thermal conductivity thus explaining the wide range of thermal conductivity reported, from 2 to 650 W/mK (and even more). See some reviews like J. Mater. Sci 2019 54:7397 by Kumanek and Janas for instance. Additionally, benchmarking thermal conductivity values reported here with some data already available for buckypapers would help put some context for the authors’ contribution.
Minor comments:
likely typo: l.2 Abstract cites CFRP low thermal conductivity while introduction l.27 cites CFRP low thermal resistance. Both can’t be true.
SEM is not really suitable to measure CNT diameters since it’s not possible to distinguish between a single CNT or CNT bundle, TEM is the most appropriate technique. Also, how many CNTs have been measured to obtain the histogram shown in Figure 2?
Author Response
Thank you for your detailed and helpful review. I will now address your comments in detail and insert the corresponding revised sections. To provide clarity, I have also highlighted the changes in the text. I hope this meets your expectations.
Comments 1: My main comment is that buckypapers are expected to exhibit highly anisotropic thermal conductivity it this does not seem to be addressed too much. In fact, in-plane thermal conductivity can easily be 1 order of magnitude higher than through-plane thermal conductivity. Although the authors focus on through-plane thermal conductivity, it would be interesting to test in-plane or lateral thermal conductivity as well. Some 3D design, or even just something simple with “buckypaper vertical walls” at the edge of the materials could drastically increase “in plane” (i.e. vertical) thermal conductivity.
Response 1:
You are absolutely right. The anisotropy of the buckypapers significantly determines the properties of the composite material. We particularly accounted for this in the visualization of the infiltration process (Figure 3) and the analysis of the electrical properties (Figure 7). A comparison between Figures 7 (b1) and (b2) reveals that the electrical conductivity at the surface (lateral direction) improves by approximately one order of magnitude more than in the bulk (vertical direction).
In these materials, electrical and thermal properties are often closely correlated. This means that a good electrical conductor is frequently also a good thermal conductor. As you correctly pointed out, the thermal conductivity in the lateral direction is therefore most likely higher than in the vertical direction.
Unfortunately, we were not able to determine this value using the light flash method. However, since lateral thermal conductivity plays a critical role in heat dissipation under one-sided thermal loading, this development should not remain unaddressed. For this reason, we have added the following paragraph:
Line 376: An even higher thermal conductivity is expected exclusively in the longitudinal direction of the sample, owing to the anisotropic structure of the carbon fibers and nanotubes. In this direction, a highly ordered, graphitic crystal structure with covalent bonds facilitates particularly efficient heat transport along direct pathways.
I find your approach of integrating the buckypapers either in a three-dimensional arrangement within the material or along the edges to enhance vertical conductivity very interesting. We will discuss this idea within our research group. It may offer a promising route to further improve the thermal resistance of the composite. I have included this line of thought in the outlook section:
Line 731: Additionally, the thermal performance of the nanocomposites could be further enhanced by incorporating buckypapers in a three-dimensional configuration or by strategically positioning them along the edges of the laminate. This strategy aims to significantly improve thermal conductivity in both the lateral and vertical directions. As a result, heat could be dissipated even more efficiently under thermal loading.
Comments 2: The introduction could be improved citing the various parameters (CNTs diameter, length, number of layers (or layer thickness), layer compression...) that have huge influence on buckypaper thermal conductivity thus explaining the wide range of thermal conductivity reported, from 2 to 650 W/mK (and even more). See some reviews like J. Mater. Sci 2019 54:7397 by Kumanek and Janas for instance. Additionally, benchmarking thermal conductivity values reported here with some data already available for buckypapers would help put some context for the authors’ contribution.
Response 2:
This is a very valuable suggestion. The introduction has been revised to highlight the various factors that influence the thermal conductivity of CNTs and buckypapers. This provides a clear justification for the wide range of reported values—ranging from 0.1 to 6600 W/mK for CNTs and from 0.1 to 770 W/mK for buckypapers. These values are supported by the review article by Kumanek and Janas, as well as three additional publications.
The calculated thermal conductivities of our buckypapers, ranging from 5 to 7 W/mK, are presented in the results section (Table 1) and can now serve as a benchmark for comparison with values reported in the introduction. The corresponding revisions are as follows:
Line 32: To mitigate the risk of localized overheating, nanoparticulate additives such as carbon nanotubes (CNTs) can be incorporated in the material due to their exceptional potential for high longitudinal thermal conductivity. However, their effective conductivity strongly depends on factors such as structure, morphology, and synthesis method. Consequently, thermal conductivity can vary widely—from below 0.1 W/(m·K) for aggregated multi-walled CNTs up to 6600 W/(m·K) for individual single-walled CNTs at room temperature.
Line 58: One promising approach to overcome this limitation is the use of pre-fabricated, thin, dense, yet flexible nanotube network films, known as buckypapers. On the one hand, these films can exhibit effective thermal conductivities ranging from 0.1 up to 770 W/mK, depending on the intrinsic properties of the CNTs, packing density, alignment, and layer structure. On the other hand, they can be directly processed with prepreg plies, enabling an innovative approach for more efficient integration of CNTs into the composite material. As a result, high particle loadings can be incorporated in a targeted and controlled manner without requiring dispersion during processing.
Comments 3: likely typo: l.2 Abstract cites CFRP low thermal conductivity while introduction l.27 cites CFRP low thermal resistance. Both can’t be true.
Response 3: This phrasing is unfortunate. In this context, low thermal resistance actually means low thermal stability. To avoid any misunderstandings, the wording has been revised as follows:
Line 27: However, compared to alternative lightweight metals, these composites are limited by the relatively low thermal stability of their polymer matrix.
Comments 4: SEM is not really suitable to measure CNT diameters since it’s not possible to distinguish between a single CNT or CNT bundle, TEM is the most appropriate technique. Also, how many CNTs have been measured to obtain the histogram shown in Figure 2?
Response 4: As you mentioned, TEM is naturally the better method for determining the exact diameter of the CNTs. However, for this work, the precise size is not crucial; rather, obtaining a rough size range and distribution is more important to assess their performance. The histogram in Figure 2 (b) comprises a total of 207 CNTs. To allow evaluation of the size distribution’s accuracy, this sample size is explicitly stated in the histogram.

Reviewer 2 Report
Comments and Suggestions for Authors
This manuscript presents primarily experimental results on the changes in composite properties as a consequence of incorporating various amounts and configurations of CNT-paper. The results are of practical importance, but the novelty is somewhat limited. Similar to numerous literature studies, the results show that basic composite properties (ILSS, tensile, compressive strength) all decrease with the addition of increasing amounts of CNTs while providing some enhancement to thermal properties. Further, most studies completely side-step the issue of added cost of CNTs. Authors need to document the added material costs as well as the processing costs due to CNTs. Authors should also address the following comments in their revision.
Authors’ choice of 10% reduction being acceptable is rather arbitrary; the shaded regions must be deleted. For instance, in Figure 14, experimental error bars associated with each ILSS data point should be noted together with an inference whether the differences are statistically significant. This can explain discrepancy such as with bop0 data that shows slight increase initially in ILSS as irradiation time increases from 0 to 5 seconds. Also, this unmodified composite shows best performance at 25 s irradiation time. Why?
Author Response
Thank you for your detailed and helpful review. I will now address your comments in detail and insert the corresponding revised sections. To provide clarity, I have also highlighted the changes in the text. I hope this meets your expectations.
Comments 1: This manuscript presents primarily experimental results on the changes in composite properties as a consequence of incorporating various amounts and configurations of CNT-paper. The results are of practical importance, but the novelty is somewhat limited. Similar to numerous literature studies, the results show that basic composite properties (ILSS, tensile, compressive strength) all decrease with the addition of increasing amounts of CNTs while providing some enhancement to thermal properties. Further, most studies completely side-step the issue of added cost of CNTs. Authors need to document the added material costs as well as the processing costs due to CNTs. Authors should also address the following comments in their revision.
Response 1: You are correct that the fundamental concept of incorporating CNTs into composite materials has existed since the early 2000s. In many studies, CNTs in powder form have been integrated into polymer melts and then wet-laminated with fabric plies. However, only low mass fractions of up to about 2% CNTs can be processed due to their high aspect ratio, which severely limits the potential for improvement.
In contrast, this work integrates CNTs in the form of buckypapers between prepreg plies to enable the incorporation of larger amounts of CNTs. The novelty of this paper lies in the detailed investigation of how the number and positioning of buckypapers within the CFRP affect processability, material properties, and thermal resistance.
I agree with you. The potential for improvement must, of course, be considered in relation to the costs, especially for large-scale applications. High-quality carbon nanotubes are already very expensive, and the additional processing into buckypapers incurs further costs. I have incorporated these remarks in the outlook section:
Line 714: Based on these findings, the comparison of various manufacturing methods reveals that the melt process is both time-consuming and requires considerable expertise. The major disadvantage, however, is that only low nanotube fractions up to 2 wt.~\% can be homogeneously incorporated due to their poor dispersibility, caused by the high aspect ratio. In contrast, buckypapers offer a simpler and faster alternative, allowing the direct incorporation of up to 8 wt.~\% nanotubes without the need for dispersion. As a result, thermal resistance can be improved by nearly threefold. However, these performance gains come at a cost: high-quality carbon nanotubes are already expensive, and the additional processing steps required to fabricate buckypapers further increase the overall material costs. Therefore, the improvements in thermal stability must be carefully weighed against economic feasibility, especially for large-scale applications.
Comments 2: Authors’ choice of 10% reduction being acceptable is rather arbitrary; the shaded regions must be deleted. For instance, in Figure 14, experimental error bars associated with each ILSS data point should be noted together with an inference whether the differences are statistically significant. This can explain discrepancy such as with bop0 data that shows slight increase initially in ILSS as irradiation time increases from 0 to 5 seconds. Also, this unmodified composite shows best performance at 25 s irradiation time. Why?
Response 2: The shaded areas are based on our experimental experience and represent typical variations arising from material quality, sample preparation, and testing methods. For the reader, this correlation may appear arbitrary. To avoid confusion, I have removed these thresholds from Figures 9 and 14, as they are not essential to the overall argument.
I would like to provide some clarification. To this end, I have included error bars and will explain the figure in more detail. Figure 14 (a1) shows the strength as a function of irradiation time. The initial strength of sample b0p0 is 71.2 ± 0.1 MPa. After thermal exposure for 5 s, it decreases to approximately 68.1 ± 1.0 MPa, as the polymer matrix near the surface begins to oxidize at temperatures below 150 °C. No increase in strength, as you suggested, was observed. After 25 s of exposure, the strength further decreases to about 26.3 ± 4.4 MPa due to the accumulation of thermo-induced damage, formation of structural defects, and depletion of the polymer matrix. The best performance is not observed at this exposure time but rather in the materials with five integrated buckypapers, as shown in Figure 14 (a2). I hope this explanation makes the figure clearer.
